# Validation of survey effort measures of grit and self-control in a sample of high school students

**Gema Zamarro**[1]*, **Malachi Nichols**[1], **Angela L. Duckworth**[2], **Sidney K. D'Mello**[3]

**1** Department of Education Reform, University of Arkansas, Fayetteville, Arkansas, United States of America, **2** Department of Psychology, University of Pennsylvania, Philadelphia, Pennsylvania, United States of America, **3** Institute of Cognitive Science, University of Colorado Boulder, Boulder, Colorado, United States of America

\* gzamarro@uark.edu

**Data Availability Statement:** All data collected in this study is considered confidential. Our IRB protocol at the University of Pennsylvania only allows sharing a fully de-identified dataset with collaborators for research analysis directly involved

## Abstract

Personality traits such as grit and self-control are important determinants of success in life outcomes. However, most measures of these traits, which rely on self-reports, might be biased when used for the purpose of evaluating education policies or interventions. Recent research has shown the potential of survey effort—in particular, item non-response and careless answering—as a proxy measure of these traits. The current investigation uses a dataset of high school seniors ($N = 513$) to investigate survey effort measures in relationship with teacher reports, performance task measures, high school academic outcomes, and college attendance. Our results show promise for use of survey effort as proxy measures of grit and self-control.

## Introduction

Though the importance of personality traits such as grit (passion and perseverance for long-term goals) and self-control (the ability to regulate attention, emotion, and behavior despite temptations) to life outcomes including education levels, career success, health outcomes, and criminal behavior is well-established [1], [2], [3]), researchers have struggled to find non-biased measures of these traits to be used for the purpose of evaluation of education policies and interventions [4], Further, many existing datasets lack any measures at all. As a result, research on how to support and develop these important traits is limited by an inability to measure them.

Recent literature has proposed the use of survey effort as proxy measures of grit and self-control to either supplement information obtained through self-reports, which might be affected by multiple types of bias (e.g., reference group bias and social desirability bias; see [4]), or to complement datasets that lack measures of these traits [5], [6], [7], [8], [9], [10], Why? Surveys take effort to complete. For students, in particular, surveys administered in classrooms can feel like schoolwork or homework. Therefore, by studying how much effort students put into surveys, we can obtain proxy measures of a student's grit and self-control.

in this project after receiving approval by the PI of the project (Angela Lee Duckworth), who assumed responsibility for how and where the data will be stored and analyzed.

**Funding:** Angela Duckworth and Sidney D'Mello received funding from the Bill and Melinda Gates Foundation (https://www.gatesfoundation.org/) and the Walton Family Foundation (https://www.waltonfamilyfoundation.org/) for this work. The funders did not play any role in the study design, data collection and analysis, decision to publish, or preparation of the manuscript.

**Competing interests:** The authors have declared that no competing interests exist.

Two measures, in particular, have shown promise: item non-response rates and careless answering. We define item non-response as the percentage of questions skipped by a respondent on a survey. Taking advantage of longitudinal nationally representative samples of adolescents and adults in the United States and Germany, Hedengren and Stratmann [11], found item non-response to be correlated with self-reported conscientiousness, a personality trait related to grit and self-control (one standard deviation increase in response rates was associated with a statistically significant 0.3 standard deviation increase in self-reported conscientiousness) and a significant predictor of earnings and mortality risks. Furthermore, Hitt, Trivitt, and Cheng [12] use six different longitudinal nationally representative samples of American youth to determine the relationship between the percentage of questions skipped and desirable self-reported outcomes measured in adulthood that are known to be related to individual's levels of grit, self-control, and related traits [1], They found that item non-response was a significant predictor of self-reported educational attainment and labor market outcomes, independent of available measures of cognitive ability (one standard deviation increase in item non-response was associated with completing between 0.1 and 0.3 fewer years of education).

In addition, some respondents might show low survey effort by answering randomly and carelessly [8] [9], Using two national longitudinal surveys, Hitt [6], found that careless answering, measured as the presence of haphazard and inconsistent responses, in adolescent respondents was associated with reporting fewer years of completed education and a decreased probability of high school completion, independent of cognitive ability (one standard deviation increase in careless answering was associated with an about 0.1 decrease in self-reported completed years of education and almost a two percentage points decrease in the probability of graduating from high school). Similarly, using data from a nationally representative internet panel of American adults, Zamarro et al. [10], found that repeated careless answering behavior among adults was negatively correlated with self-reported grit [13], and self-reported conscientiousness [14] (partial correlations ($r_{xy,z}$) of about -0.15 after controlling for cognitive ability and demographic information), and positively correlated with neuroticism, shedding light on its validity as a measure. They also determined that careless answering was a significant negative predictor of self-reported total years of education and lower levels of self-reported income and career success.

Although recent research has shown that survey effort measures can be promising proxy measures for personality traits related to grit and self-control [11] [12] [6] [10], these validation exercises have relied on self-reported measures of personality traits and outcome variables, and lacked external sources of information. The sole exception is the work of Hedengren and Stratmann [11], which used information on earnings and mortality risks from administrative sources. We aim to fill this gap in the literature by studying the relationship between survey effort measures and teacher evaluations of traits, performance task measures, and external outcome measures.

We use data on a sample of 513 high school seniors attending a public school in the Northeastern United States. Although our dataset is comprised of a relatively small convenience sample, it is a unique one: it collates a diverse set of measures of students' personality traits, including self-reported measures, teacher reports, performance measures from two validated tasks, and administrative records. Complementing the work of Hitt, Trivitt, and Cheng [12] and Hitt [6], we study the correlation of survey effort measures with students' self-reported measures of grit and self-control and academic outcomes at the end of high school, college attendance one year after graduation, and, more importantly, with teacher reports on these traits. Second, we study the relationship between survey effort measures and other performance task measures designed to capture related traits like academic diligence, effort put

forward by students on tedious school related tasks [15], and frustration tolerance, the ability to overcome frustration arising from challenges that block goals [16], Our results suggest that survey effort can be used as proxy measures of grit and self-control.

## Materials and methods

### Participants

This study was approved by University of Pennsylvania IRB (Protocol 814991) and the University of Arkansas IRB (Protocol 16-10-164). The data used is from a study on college persistence led by a research team at the University of Pennsylvania. In the spring of 2014, the team collected data from 513 high school seniors attending a public high school in the Northeastern United States. The research team recruited participants through opt-out parental consent forms distributed by the school administration. If the parent did not wish for the child to be part of the study, they could indicate so by signing the provided form and sending it back to the school. Alternatively, they could call or email the principal investigator of the study. In addition, non-opted out students were also given a child assent from at the beginning of the first session of the study. Through this form they got the option to also opt out of the study themselves. A total of 154 students opted out of the study. One year later, the research team used National Student Clearinghouse (a non-profit organization offering nationwide college enrollment and degree attainment data, see https://www.studentclearinghouse.org/) to track college enrollment status of as many participants as possible, which resulted in a study with adequate power to detect small to medium effects [16],

According to demographic information obtained from school records (see Table 1), 41% of students were African American, 36% White, 20% Asian, and 3% Hispanic; 54% were female. Half (51%) qualified for Free and Reduced-price Lunch (FRL).

### Assessments and measures

In a first session, during the month of November 2012, students were administered the assent forms and a vocabulary test during planning periods in school (37 minutes sessions). A large make-up session with about 300 students was held on the final day of testing in the library computer lab.

In a second session, during January 2013, students completed the Matrix Reasoning subtest of the Kaufman Brief Intelligence Test (KBIT) [17], then an online questionnaire on student autonomy, purpose for applying to college, growth mindset measures, locus of control, trust and belonging, feelings toward math, Big 5 personality questions, positive and negative affect, believe about the role of effort, and life satisfaction. Afterward students completed the Academic Diligence Task (ADT), which is described in more detail below. Finally, students answered 10 questions related to socio-economic status, participation in extracurricular activities, and description of self. This was a 2.5 hours session that took place in the cafeteria (50 to 200 students per day) or individual classrooms in school (about 30 students per class per day).

In a final session in May 2013, students completed the Mirror Tracing Frustration Task (MTFT–described below) during senior planning periods at the school library. Students were tested during four periods per day with two classes of students per period (about 30 to 60 students per period).

Separately, three teachers provided overall ratings about all their participating students' levels of grit and self-control. Participating students and teachers were compensated for their time with small, non-monetary rewards (e.g., credit to the school library coffee house). Teachers' compensation was less than $25 in value, and students' compensation was less than $5 in value.

**Table 1. Summary statistics for demographic and outcome variables.**

|  | Measure | Mean | Standard Deviation | Minimum | Maximum |
|---|---|---|---|---|---|
| **Demographic** | | | | | |
|  | Age | 17.93 | 0.53 | 16 | 21 |
|  | Female | 0.54 | 0.50 | 0 | 1 |
|  | Asian | 0.20 | 0.40 | 0 | 1 |
|  | African American | 0.41 | 0.49 | 0 | 1 |
|  | Hispanic | 0.03 | 0.16 | 0 | 1 |
|  | Caucasian | 0.36 | 0.48 | 0 | 1 |
|  | ELL | 0.14 | 0.35 | 0 | 1 |
|  | SPED | 0.14 | 0.35 | 0 | 1 |
|  | FRL | 0.51 | 0.50 | 0 | 1 |
|  | Median Household Income ($) | 52,530 | 22,915 | 9,471 | 128,618 |
|  | KBIT Scaled Score | 94.26 | 21.43 | 40 | 132 |
| **Outcome** | | | | | |
|  | HS GPA Senior | 85.07 | 7.66 | 55 | 100 |
|  | HS Graduate | 0.95 | 0.22 | 0 | 1 |
|  | End of Year Math Test | 1529.32 | 54.49 | 1363 | 1698 |
|  | End of Year Reading Test | 1528.77 | 48.32 | 1385 | 1706 |
|  | Attempted SAT | 0.51 | 0.50 | 0 | 1 |
|  | Mean SAT | 1414.80 | 254.25 | 820 | 2060 |
|  | College Enrollment for 1 Year | 0.64 | 0.48 | 0 | 1 |
|  | 4-year College Enrollment for 1 Year | 0.43 | 0.50 | 0 | 1 |
|  | 4-year College Enrollment for 1 Year (Full-Time) | 0.40 | 0.49 | 0 | 1 |

$N$ = 513 students. ELL, English Language Learner students; SPED, Special Programs Education Students; FRL, students eligible for Free or Reduced-price Lunch.

**Survey effort measures.** *Item non-response*. Following Hitt, Trivitt, and Cheng [12], we parametrized survey effort by computing two measures of survey item non-response. We determined item non-response by dividing the total number of questions left blank by the number of answerable questions, given legitimate skips, that is excluding those questions left blank because the student was not requested to answer given the routing in the survey and prior answers. Relatedly, we also computed a dichotomous item non-response measure as a binary indicator for the student leaving any answerable question blank, dependent on legitimate skips. We computed this measure because almost half of our sample (47%) completed the entire survey. Fig 1 shows the distribution of survey item non-response rates in our sample, among those who left at least one question blank.

*Careless answering*. Following Hitt [6], the second way we parametrized survey effort is through measures of careless answering. The idea behind this measure is as follows: Consider a reliable, validated scale with a number of items. If the scale is reliable, each item will consistently measure the same underlying construct. Individual responses to each item would be closely predicted by responses to other items in the same scale. Thus, we interpreted deviations in responses from predicted values given responses to other items in the scale as measures of careless answering.

In practice, we first identified reliable scales within the student survey with Cronbach's alpha reliability coefficients of at least 0.7 [18], We excluded the self-reported scales of grit and self-control used to validate survey effort measures in this paper. In total, we identified the following seven scales: a trust scale, a belonging scale, an interest in school scale, an academic self-efficacy scale, a distress tolerance scale, a purpose scale, and a brief self-control scale. For

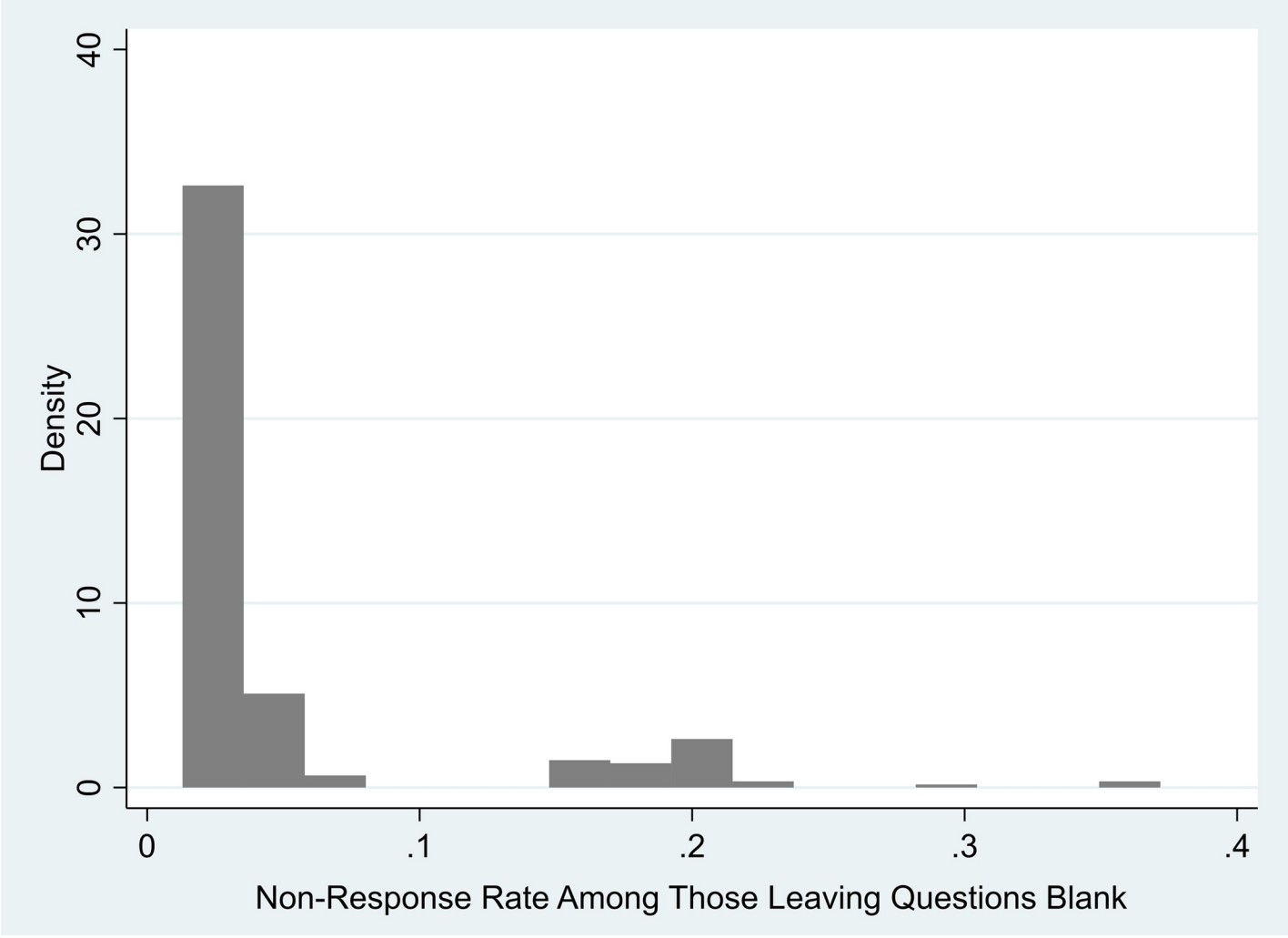

**Fig 1. Distribution of item non-response rates among those leaving questions blank.**

each, we did a regression analysis of responses for each item compared to the average score of the rest of the items in that scale. Then, we computed residuals from each of these regression models to capture the extent to which the response to a particular item is unpredictable based upon the response patterns of the individual student and others in the analytic sample. We standardized the absolute values of each of these residuals to account for any differences across the items within the same scale. We then averaged these standardized residuals within scales and standardized them again to take into account differences across scales (e.g., different total number of items, answer options). Finally, we calculated a composite careless answering score by averaging these standardized averaged residuals at the student level, with higher values of this measure indicating higher levels of carelessness or unpredictability in responding. The Table A.1 in S1 Appendix displays the Cronbach's alpha reliability coefficients for each scale we considered in our careless answering measure, the items included in each, as well as the average absolute residuals associated with each item in each scale following the regression analysis just described above.

Fig 2 shows the distribution of careless answering in our sample. Since the careless answer-ing measure is standardized by construction, mean and standard deviation are not very

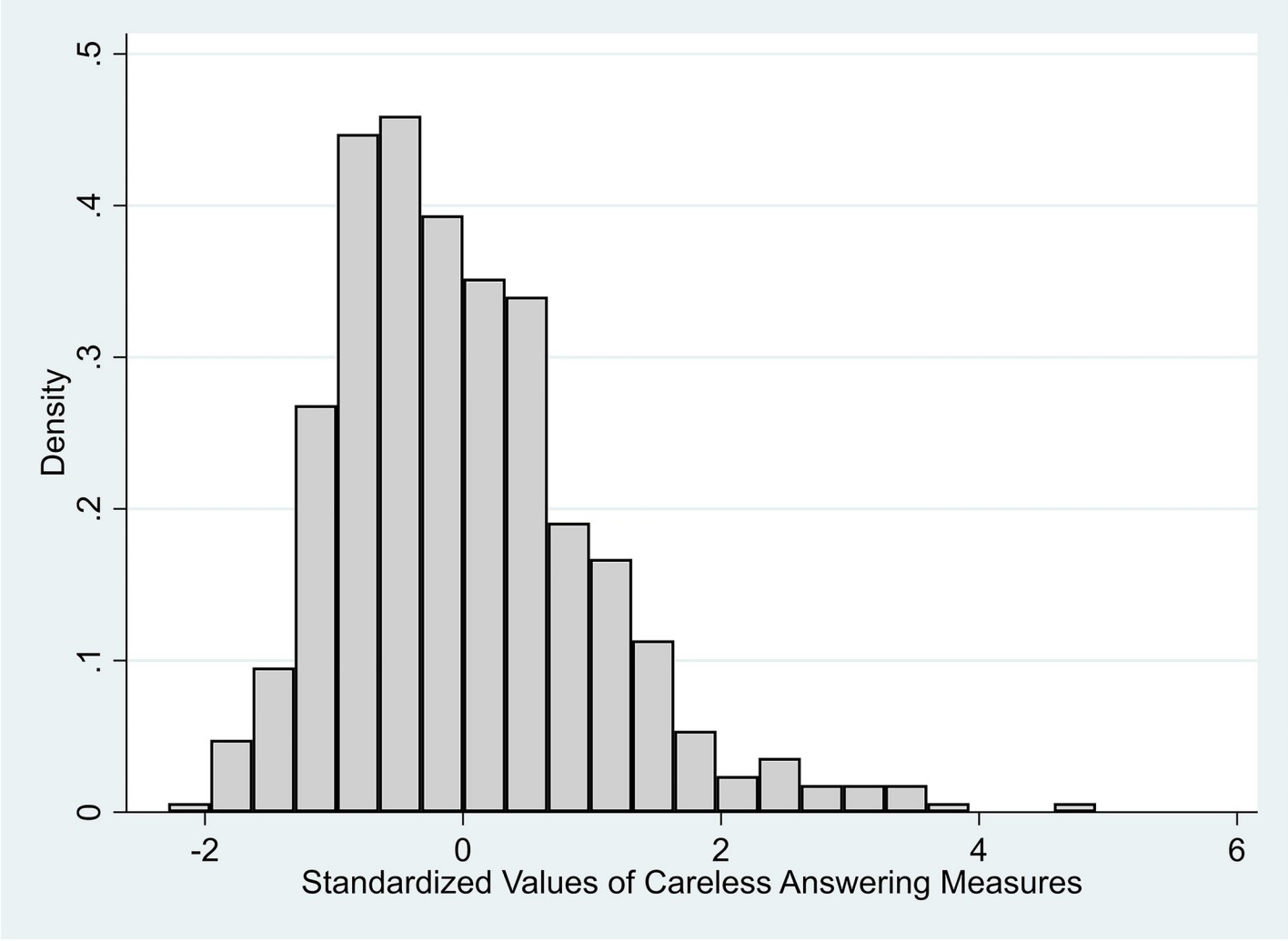

**Fig 2. Distribution of careless answering measures.**

informative. However, we note considerable variation across students in the sample as presented in the summary statistics in Table 2.

Item non-response and careless answering seem to be different approaches to exerting low survey effort. On any given item, careless answering and item non-response are mutually exclusive approaches, so it is impossible to do both at the same time. The participant-level Pearson's correlation coefficient between these two measures of survey effort is 0.17, suggesting complementary measures.

*Teacher reports*. Three teachers (homeroom, English, and social studies) provided an overall rating of each of their students participating in the study on grit and self-control and answered additional questions about classroom behavior and work ethic. To minimize burden, teachers were shown the items from the grit scale students were asked to complete [13], described in more detail below, and asked to rate how much these items as a whole described each student using a 5-point Likert-type scale. Teachers were also asked to report on students' self-control according to an 8-item Likert-type scale from the Brief Self-Control Scale [19], which students had been asked to complete, also described in detail below. This measurement approach using

**Table 2. Summary statistics for measures of character traits.**

| | Measure | M | SD | Minimum | Maximum |
|---|---|---|---|---|---|
| Survey Effort | | | | | |
| | Item Non-Response (%) | 2.41 | 5.35 | 0.00 | 37.18 |
| | Dichotomous Item Non-Response | 0.53 | 0.50 | 0.00 | 1.00 |
| | Careless Answering | 0.00 | 1.00 | -2.24 | 3.93 |
| Performance Task Measures | | | | | |
| | Diligence Task, Percentage Time Spent on Math | 0.64 | 0.30 | 0.00 | 1.00 |
| | Frustration Task, Percentage Time Spent Tracing | 0.55 | 0.27 | 0.00 | 1.00 |
| Self-Reported Measures | | | | | |
| | Grit | 3.76 | 0.71 | 1.00 | 5.00 |
| | Locus of Control | 4.57 | 0.75 | 2.50 | 6.00 |
| | Self-Control Combined (Work and Interpersonal) | 3.61 | 0.60 | 1.00 | 5.00 |
| Teacher-Reported Measures | | | | | |
| | Work Self-Control | 3.72 | 0.88 | 1.00 | 5.00 |
| | Interpersonal Self-Control | 4.21 | 0.77 | 1.00 | 5.00 |
| | Grit | 3.53 | 0.87 | 1.00 | 5.00 |
| | Redirection | 0.92 | 1.16 | 0.00 | 5.00 |
| | Homework Completion | 77.69 | 21.63 | 0.00 | 100.00 |

$N$ = 513 students. The statistics reported for the Frustration Task are from an analytical sample of $n$ = 391. Following Meindl et al. [16], we removed participants if they failed to complete a practice trial preceding the actual task, fully completed tracing the shape, experienced technical problems within the task, or were not allowed an adequate amount of time to complete the task.

a single overall assessment of a personality traits has been proven to show adequate levels of convergent and discriminatory validity, test-retest reliability, and convergence between self- and observed-ratings [20] [21], Since three teachers reported on each child, individual z-scores were averaged for individual students, giving each student a unique construct score to increase validity [22], High scores represent higher levels of that trait.

To measure classroom behavior and work ethic, we asked teachers to report on students' redirection (reminders to stay on task or follow rules) and homework completion. For redirection, teachers estimated the number of times the student required redirection within the last week, with options ranging from 0 to 5 or more times. The three teachers' scores were averaged to give each student a redirection score. A high number of redirects could represent a lack of diligence amongst other factors. Finally, we asked teachers to rate students on homework completion, giving the percentage of assignments (from 0 to 100) the student completed on time and received a passing grade. The three teachers' scores were averaged to give each student a homework completion score. A higher percentage infers that the student has high levels of work ethic.

**Direct performance task measures.** *Academic Diligence Task (ADT).* A computer-generated task designed to measure academic diligence [15], called the ADT gives students the option to either perform simple math problems, after being told about the benefits of this type of exercises, or consume media by watching online video clips or playing online games. We measured academic diligence by the percentage of the total task time (12 minutes) a student spent completing math problems instead of consuming media. Higher percentages represent higher levels of academic diligence.

In a sample of over 900 high school students, Galla et al. [15], found that measures of student engagement in the ADT were correlated with self-reported measures of conscientiousness ($r_{xy,z}$ = 0.09), self-control ($r_{xy,z}$ = 0.15), and grit ($r_{xy,z}$ = 0.17). Performance on the ADT was

also predictive of the student's high school Grade Point Average (GPA), standardized test scores, high school graduation, and college enrollment, even after controlling for potential confounds including cognitive ability and sociodemographic characteristics.

*Mirror Tracing Frustration Task (MTFT)*. Participants were also asked to complete the Mirror Tracing Frustration Task (MTFT) [16], The MTFT measures frustration tolerance. During this task, students were given the option to trace a shape using the mouse on their computer or consume media by watching online videos. However, using the mouse produced movements in the opposite direction. There was also a random drift added to each mouse moment, so perfect control was not possible. This required students to use high levels of concentration when performing the task and induced frustration. If the student stopped tracing or traced off the shape, the task automatically restarted. Students were informed about the importance of developing perceptual-motor skills for various real-world tasks in order to motivate the tracing task, but had the option to switch between the task and media as often as they desired. Frustration tolerance was measured as the percentage of the total task assigned time (5 minutes) a student spent tracing. Using this same data, Meindl and colleagues [16] showed that higher frustration tolerance was significantly associated with self-reported and teacher-reported grit and self-control measures ($r_S$ = 0.11 to 0.22), as well as high school GPA, standardized test scores, and college persistence.

**Self-reported measures.** We also study the relationship between survey effort measures and the following self-reported measures collected in the study.

*Grit*. Following Duckworth and Quinn [13], students were asked to rate how true five statements described themselves on a 5-point Likert-type scale from 1 = *not at all true* to 5 = *completely true*. These statements included, for example, "I finish whatever I begin" and "I stay committed to my goals." We averaged each student's item scores to create a grit score for each respondent. Possible grit scores range from one to five, with a high score representing high values of grit. This scale showed high reliability in our sample with a Cronbach's alpha of 0.8.

*Self-control*. Students were also asked to complete eight items from the Brief Self-Control Scale [19], This scale consisted of a combination of four questions pertaining to schoolwork and four questions pertaining to interpersonal situations. Students then rated how true the eight statements were for themselves using a 5-point Likert-type scale from 1 = *not at all true* to 5 = *completely true*. The statements for work skills included "I come to class prepared" and "I get to work right away, instead of waiting until the last minute," while the statements relating to interpersonal skills included "I allow others to speak without interruption" and "I control my temper." Scores from each scale were averaged to create a combined self-control score for each student. Average scores were also computed separately to represent self-control in work and self-control related to interpersonal skills. Scores ranged from 1 to 5 with a high score meaning the student has high levels of self-control. The combined self-control scale showed high reliability in our sample with a Cronbach's alpha of 0.8.

*Locus of control*. Finally, students were also asked to complete a 4 items locus of control scale [23] using a 6-point reporting scale. This scale captures how strongly students believe they have control over the situations and experiences that affect their educational outcomes. The items in this scale include, for example, "Getting good grades is a matter of luck" and "If you get bad grades, it's not your fault."

**Outcome measures.** We also studied the relationship between survey effort measures and other outcome variables to further study the criterion validity of survey effort measures of grit and self-control. For this purpose, our outcome measures included high school GPA ranging from 0 to 100 (In the state where our data comes from, high schools vary on their calculations and scale used for their GPAs. Therefore, we converted GPA to a 100-point scale with the help of district provided handbooks and information from the College Board), a binary variable

indicating if the student graduated high school, a binary variable indicating if the student attempted to take the Scholastic Aptitude Test (SAT), the total SAT score in the first attempt (for those who attempted the test) ranging from 600 to 2400, which was the sum total of the critical reading, math, and writing scores. Furthermore, we constructed three binary variables indicating if the student was continuously enrolled in college for one year after graduating high school, if the institution was a four-year college, and if the student was continuously enrolled full-time in that four-year college. Additionally, we also studied the relationship between survey effort measures and performance on students' final senior year assessments in math and reading, which are part of the state's high school graduation requirements; scores ranged from 1200 to 1800.

**Cognitive ability and other data.** To control for cognitive ability, we used measures of students' performance on the matrix reasoning subset of the Kaufman Brief Intelligence Test (KBIT) [17], scaled scores ranged from 40 to 132. Our analysis also includes controls for age, gender, ethnicity, English Language Learner (ELL), Special Education Status (SPED), free and reduced-price lunch (FRL) status, and household income.

**Empirical strategy for validation of measures.** For survey effort measures to be valid proxy measures of grit and self-control, they should be correlated with other measures of these character skills (convergent validity) as well as with other outcome variables known to be correlated with the same latent skills (criterion validity). Accordingly, we computed Spearman correlations and partial rank correlations (controlling for cognitive ability and socio-demographic information) for our measures of survey effort (i.e,. non-response rates and measures of careless answering) with self- and teacher- reported grit and self-control, with an expectation that the correlations would be negative. We expected to find negative correlations of both survey effort measures and teacher-reported homework completion, and positive correlations with redirection. Finally, we expected to find negative correlations between survey effort and diligence and frustration tolerance as measured through the relevant performance tasks.

The last set of analyses looked at criterion validity of item non-response and careless answering measures of survey effort. To do so, we estimated linear regression models and linear probability models to predict (from survey effort measures) each of the following academic outcomes: high school GPA, high school graduation, attempt to take the SAT, SAT scores if attempted, end-of-senior-year math and reading test scores, college enrollment in the first year after high school graduation, enrollment in a four-year college, and full-time enrollment in a four-year college. For binary outcomes, we also estimated discrete choice logit models. Results were similar to those in the linear probability models presented here. We estimated separate models for item non-response rates, dichotomous non-response, and careless answering measures as specified below:

$$Academic\ Outcome_i = \beta_0 + \beta_1 Survey\ Effort_i + \beta_2 Cognitive\ Ability_i + \beta_3 X_i + \varepsilon_i \tag{1}$$

Our models controlled for cognitive ability using the KBIT scaled score. $X_i$ represents a vector of student socio-demographic controls, including age, ethnicity, gender, English Language Learner (ELL) status, Free-Reduced Price Lunch (FRL) status, Special Education (SPED) status, and parental income. We reported estimated coefficients along with standardized regression coefficients for all models. For comparison, we also estimated models including direct performance task measures of academic diligence and frustration tolerance, as well as teacher reported and self-reported related measures.

## Results

### Descriptive statistics

Table 1 includes summary statistics for our outcome variables. On a scale of 0 to 100, the students had an average high school GPA of 85, and 95% of our sample of high school seniors graduated high school. Only half of the sample, however, attempted to take the SAT, but about 60% enrolled in college after graduation. Of these, 43% enrolled in a four-year college and 40% did so full time.

On average, students did not answer 2% of the items they were asked to complete (see Table 2). Forty-seven percent of students answered all the questions in the survey. These item non-response rates are similar to those found by Hitt, Trivitt, and Cheng [12] in multiple nationally representative samples of adolescents. Our careless answering measure, which captures inconsistent responses, ranges from -2.2 to 3.9. This indicates considerable variation in the degree of care that students put into completing the surveys, with some being more careful than the average (negative values) and some being less careful (positive values). For the performance task measures, students devoted an average of 64% of the assigned time (about 10 minutes) engaged in the math exercises in the diligence task. They spent an average of 55% of the assigned time (almost three minutes) tracing instead of engaging with the distractors when completing the frustration task.

The average self-reported grit of students in our sample was almost 4 (out of 5). Similarly, the students scored an average of almost 4 on the self-control combined scale and about 4.6 on the locus of control scale. Additionally, teachers reported an average of 3.5 in the level of grit of students in our sample, 3.7 in the level of work-related self-control and 4.2 in the level of interpersonal self-control. Teachers reported that students needed redirection on average about once during the previous week and they completed, on average, about 78% of the assigned homework on time and with a passing grade.

### Relationship among character trait measures

Table 3 presents Spearman's correlations among our proposed survey effort measures and student self-reported and teacher-reported measures of character traits. As expected, item non-response rates and careless answering were negatively correlated with self-reported grit and self-control as well as teacher-reported grit, self-control, and homework completion. Additionally, both survey effort measures were positively correlated with teacher redirection. Importantly, they were both negatively correlated with performance on both the diligence and frustration tasks, which corresponds with what we expected (i.e., lower levels of effort on the survey correspond with lower levels of performance in these tasks).

Table 3 also shows partial rank correlations among these measures after controlling for students' cognitive ability and socio-demographic information. We observed a similar pattern compared to the zero-order correlations, but partial correlations with teacher reports and performance task measures were smaller. Although the magnitudes of the correlations between survey effort and survey self-reported measures may appear small, they are at least as large as the correlations reported in prior literature validating other behavioral-task measures of conscientiousness, grit, and self-control [24] [15] [16].

### Relationship between survey effort measures and academic outcomes

We find evidence of criterion validity with respect to the predictive power of survey effort measures on high school and college academic outcomes. For comparison, we also examined the predictive power of the performance task measures, teacher reported, and self-reported

**Table 3. Spearman and partial rank correlations between performance task measures, self-reports, and teacher reports.**

| | | Item Non-response | | Careless Answering | |
|---|---|---|---|---|---|
| | | (1)[b] | (2)[c] | (1)[b] | (2)[c] |
| **Self-Reported Measures** | | | | | |
| | Grit | -0.118* | -0.155* | -0.024 | -0.066 |
| | Locus of Control | -0.093* | -0.091 | 0.038 | -0.022 |
| | Self-Control Combined | -0.135* | -0.148* | -0.104* | -0.114* |
| | Self-Control Work | -0.081* | -0.101* | -0.127* | -0.153* |
| | Self-Control Interpersonal | -0.144* | -0.155* | -0.042 | -0.035 |
| **Teachers-Reported Measures** | | | | | |
| | Teacher-Reported Grit | -0.216* | -0.184* | -0.170* | -0.131* |
| | Teacher-Reported Work Self-Control | -0.201* | -0.164* | -0.165* | -0.107* |
| | Teacher-Reported Interpersonal Self-Control | -0.147* | -0.122* | -0.092* | -0.065 |
| | Teacher-Reported Redirection | 0.112* | 0.111* | 0.133* | 0.091 |
| | Teacher-Reported HW Completion | -0.157* | -0.122* | -0.105* | -0.058 |
| **Performance Task Measures** | | | | | |
| | Diligence Task PT Math | -0.152* | -0.084 | -0.163* | -0.125* |
| | Frustration Task PT Trace[a] | -0.104* | -0.067 | -0.134* | -0.102* |

* represents p-value < 0.05. Total sample of 513 students.

[a] The statistics reported for the Frustration Task are from a sample of 391 students.

[b] corresponds to Spearman correlations

[c] corresponds to partial correlations controlling for KBIT Scaled Score, Age, Ethnicity, Gender, FRL, SPED, ELL, and household income.

related measures. Table 4 presents the results of linear regression models for student academic outcomes, following the specifications described above in Eq (1), when different survey effort measures and performance task measures were included as explanatory variables. Regressions that use SAT scores as a dependent variable were limited to only those students who attempted the SAT. Sample sizes varied depending on the available information for each individual regression model, ranging from 392 to 458 observations and from 216 to 240 for SAT score models. Similarly, following Meindl et al. [16], results for the frustration task excluded data from students if they failed to complete a practice trial preceding the actual task, fully completed tracing the shape, experienced technical problems during the task, or were not allowed an adequate amount of time to complete the task due to data collection constraints. As a robustness check, we also performed estimates including the full data set (i.e,. N = 513) and the main results were comparable to the ones presented above.

In the results for survey effort measures, we found that a standard deviation increase in item non-response led to an almost 0.2 standard deviation decrease in high school GPA, a 0.2 standard deviations decrease in the probability of attempting the SAT, a 0.14 decrease in SAT scores if attempted, an almost 0.2 standard deviation decrease in end-of-senior-year math and reading scores, and a 0.2 standard deviation decrease in the probability of being enrolled in college one year after graduation, keeping cognitive ability and demographic information fixed (see Table 4). We also estimated models that include both item non-response rates and a binary indicator for leaving any question blank to see if these behaviors were related to academic outcomes. We found that this was generally the case: both were significant predictors of these academic outcomes. Similarly, a one standard deviation increase in careless answering was associated with a 0.12 standard deviation decrease in GPA, a comparable decrease in the probability of attempting the SAT, about 0.1 standard deviation decrease on end-of-senior-year math and reading exams, and a 0.08 standard deviation decrease in the probability of

**Table 4. Estimated coefficients of linear regression models predicting academic outcomes.**

|  | High School GPA | High School Graduation | Attempt SAT | SAT | End of Year Math | End of Year Read | College Enroll 1 year | 4yr College Enroll 1 year | 4yr College Enroll Full Time 1 year |
|---|---|---|---|---|---|---|---|---|---|
| **Item Non-Response (%)** | -0.271*** | 0.0007 | -0.021*** | -12.116** | -1.917*** | -1.848*** | -0.020*** | -0.019*** | -0.017*** |
|  | [-0.196] (0.060) | [0.024] (0.001) | [-0.236] (0.004) | [-0.139] (4.956) | [-0.193] (0.394) | [-0.197] (0.388) | [-0.238] (0.004) | [-0.213] (0.004) | -0.192 (0.004) |
| *Adj R-squared* | 0.240 | 0.077 | 0.161 | 0.273 | 0.374 | 0.316 | 0.126 | 0.167 | 0.159 |
| **Dichotomous Item Non-response** | -2.220*** | -0.021 | -0.271*** | -64.868** | -17.890*** | -22.212*** | -0.239*** | -0.241*** | -0.212*** |
|  | [-0.144] (0.664) | [-0.065] (0.015) | [0.271] (0.044) | [-0.125] (29.650) | -0.164 (4.311) | [-0.228] (3.993) | [-0.249] (0.043) | [-0.242] (0.044) | [-0.214] (0.044) |
| *Adj R-squared* | 0.224 | 0.080 | 0.178 | 0.270 | 0.365 | 0.329 | 0.132 | 0.180 | 0.168 |
| **Dichotomous Item Non-response** | -1.195* | -0.029* | -0.210*** | -40.470 | -11.218*** | -17.194*** | -0.175*** | -0.186*** | -0.162*** |
|  | [-0.078] (0.719) | [-0.090] (0.016) | [-0.209] (0.048) | [-0.078] (33.11) | [-0.103] (4.670) | [-0.176] (4.360) | [-0.183] (0.047) | [-0.186] (0.047) | [-0.164] (0.047) |
| *Adj R-squared* | 0.243 | 0.081 | 0.194 | 0.275 | 0.381 | 0.340 | 0.151 | 0.193 | 0.179 |
| **Item Non-Response (%)** | -0.227*** | 0.002 | -0.014*** | -9.053 | -1.491*** | -1.156*** | -0.014*** | -0.012** | -0.011** |
|  | [-0.163] (0.065) | [0.061] (0.001) | [-0.151] (0.004) | [-0.104] (5.548) | [-0.150] (0.430) | [-0.123] (0.420) | [-0.163] (0.004) | [-0.137] (0.004) | [-0.125] (0.004) |
| *Adj R-squared* | 0.243 | 0.081 | 0.194 | 0.275 | 0.381 | 0.340 | 0.151 | 0.193 | 0.179 |
| **Careless Answering** | -1.967*** | -0.007 | -0.131*** | 19.536 | -15.749*** | -10.331** | -0.085** | -0.062 | -0.050 |
|  | [-0.119] (0.724) | [-0.021] (0.016) | [-0.122] (0.049) | [0.033] (34.598) | [-0.133] (4.781) | [-0.097] (4.535) | [-0.083] (0.048) | [0.058] (0.049) | [-0.047] (0.048) |
| *Adj R-squared* | 0.217 | 0.076 | 0.122 | 0.255 | 0.355 | 0.287 | 0.079 | 0.127 | 0.126 |
| **Diligence Task PT Math** | 3.708*** | -0.012 | 0.039 | 96.553* | 25.956*** | 20.649*** | 0.1485* | 0.108 | 0.094 |
|  | [0.145] (1.207) | [-0.021] (0.028) | [0.023] (0.084) | [0.114] (53.832) | [0.144] (7.962) | [0.126] (7.513) | [0.093] (0.081) | [0.064] (0.082) | [0.057] (0.082) |
| *Adj R-squared* | 0.232 | 0.077 | 0.110 | 0.242 | 0.352 | 0.304 | 0.091 | 0.147 | 0.127 |
| **Frustration Task PT Tracing** | 3.707*** | 0.030 | 0.214** | 41.058 | 34.872*** | 20.138** | 0.145 | 0.119 | 0.098 |
|  | [0.132] (1.414) | [0.052] (0.030) | [0.118] (0.096) | 0.046 (61.836) | [0.196] (8.702) | [0.117] (8.684) | [0.086] (0.092) | [0.065] (0.098) | [0.054] (0.097) |
| *Adj R-squared* | 0.208 | 0.114 | 0.117 | 0.171 | 0.278 | 0.223 | 0.068 | 0.095 | 0.100 |

Standardized coefficients in brackets. Standard errors of estimated coefficients in parenthesis. Additional controls included in the model are: KBIT Scaled Score, Age, Ethnicity, Gender, FRL, SPED, ELL and household income.

* Indicates P-values<0.1

** Indicates P-values<0.05, and

*** Indicates P-values<0.01.

being enrolled in college one year after graduation, all else being equal. Finally, generally none of the survey effort measures was found to be a predictor of high school graduation, only dichotomous item non-response appears marginally significant when included along with item non-response rates. This result could be because a great majority of students in our sample (95 percent) graduated high school.

We found that both the academic diligence and frustration tasks significantly predicted GPA and end-of-senior-year math and reading test scores. Estimated effects were comparable in size to those we found for survey effort measures. One standard deviation increase in performance in the diligence task is associated with a 0.14 standard deviation increase in GPA, a 0.11 standard deviation increase in SAT scores, and about 0.14 standard deviation increase in end-of-senior-year math and reading test scores. Performance in the diligence task also significantly predicted SAT scores and college enrollment, but only marginally. Finally, performance on the frustration task significantly predicted the probability of attempting the SAT. These findings confirm the work of Galla et al. [15] and Meindl et al. [16] who found that

**Table 5. Estimated coefficients of linear regression models predicting academic outcomes.**

| | High School GPA | High School Graduation | Attempt SAT | SAT | End of Year Math | End of Year Read | College Enroll 1 year | 4yr College Enroll 1 year | 4yr College Enroll Full Time 1 year |
|---|---|---|---|---|---|---|---|---|---|
| **Teacher-Reported Grit** | 4.648*** [0.505] | 0.037*** [0.206] | 0.146*** [0.253] | 29.751 [0.090] | 17.450*** [0.274] | 16.976*** [0.295] | 0.141*** [0.257] | 0.116*** [0.202] | 0.115*** [0.202] |
| | (0.334) | (0.008) | (0.026) | (18.957) | (2.464) | (2.320) | (0.025) | (0.026) | (0.025) |
| *Adj R-squared* | 0.449 | 0.093 | 0.166 | 0.262 | 0.407 | 0.362 | 0.132 | 0.161 | 0.162 |
| **Teacher Reported work self-control** | 4.588*** [0.500] | 0.040*** [0.230] | 0.161*** [0.283] | 25.883 [0.078] | 15.435*** [0.244] | 16.126*** [0.288] | 0.132*** [0.243] | 0.118*** [0.208] | 0.120*** [0.215] |
| | (0.341) | (0.008) | (0.025) | (19.410) | (2.528) | (2.312) | (0.025) | (0.026) | (0.025) |
| *Adj R-squared* | 0.438 | 0.101 | 0.179 | 0.260 | 0.390 | 0.355 | 0.123 | 0.162 | 0.165 |
| **Teacher Reported Interpersonal self-control** | 2.768*** [0.262] | 0.030*** [0.150] | 0.126*** [0.193] | 70.297*** [0.172] | 12.181*** [0.169] | 15.042*** [0.231] | 0.127*** [0.203] | 0.128*** [0.197] | 0.120*** [0.186] |
| | (0.454) | (0.009) | (0.030) | (23.502) | (2.941) | (2.756) | (0.029) | (0.030) | (0.030) |
| *Adj R-squared* | 0.266 | 0.073 | 0.140 | 0.283 | 0.362 | 0.327 | 0.107 | 0.158 | 0.154 |
| **Teacher Reported Redirection** | -2.575*** [-0.381] | -0.016** [-0.123] | -0.086*** [-0.201] | -11.119 [-0.044] | -8.558*** [-0.182] | -10.410*** [-0.252] | -0.063*** [-0.154] | -0.057*** [-0.134] | -0.050*** [-0.119] |
| | (0.270) | (0.006) | (0.019) | (14.37) | (1.900) | (1.718) | (0.019) | (0.019) | (0.019) |
| *Adj R-squared* | 0.341 | 0.067 | 0.142 | 0.254 | 0.367 | 0.338 | 0.091 | 0.141 | 0.138 |
| **Teacher Reported Homework Completion** | 0.107*** [0.289] | 0.001*** [0.194] | 0.004*** [0.172] | -0.762 [-0.055] | 0.313*** [0.120] | 0.253** [0.106] | 0.005*** [0.209] | 0.003*** [0.120] | 0.002** [0.111] |
| | (0.015) | (0.0003) | (0.001) | (0.804) | (0.107) | (0.103) | (0.001) | (0.001) | (0.001) |
| *Adj R-squared* | 0.284 | 0.092 | 0.130 | 0.255 | 0.338 | 0.286 | 0.106 | 0.134 | 0.134 |

Standardized coefficients in brackets. Standard errors of estimated coefficients in parenthesis. Additional controls included in the model are: KBIT Scaled Score, Age, Ethnicity, Gender, FRL, SPED, ELL and household income.

* Indicates P-values<0.1

** Indicates P-values<0.05, and

*** Indicates P-values<0.01.

performance on the academic diligence task and the frustration task predicted high school academic outcomes and college enrollment.

Similarly, for comparison, Tables 5 and 6 show the predictive power of teacher reported and student self-reported related measures. Student self-reported measures of grit and self-control appear to be comparable predictors than survey effort measures but teacher reports appear to be a better predictor of student academic outcomes than survey effort or performance tasks. We find that all types of teacher reports considered significantly predict senior year GPA, the probability of high school graduation, attempting the SAT, performance in the Keystone reading and math tests and college enrollment. Finally, the only significant predictor of SAT scores, among those who took the test, is teacher reported interpersonal self-control. Effect sizes are also generally larger than those found for survey effort or performance tasks. It should be stressed, however, that as it was the case with self-reports, teacher reports are subject to similar biases and manipulation if used for evaluation purposes. Also, they are often not available in researcher's datasets. However, when available they seem to be good measures of students' character traits. Survey effort measures, on the other hand, still showed predictive power and concurrent validity and so, are potentially a good proxy measure of grit and self-control when other measures are not available or when we suspect might be affected by manipulation or other sources of bias.

**Table 6. Estimated coefficients of linear regression models predicting academic outcomes.**

| | High School GPA | High School Graduation | Attempt SAT | SAT | End of Year Math | End of Year Read | College Enroll 1 year | 4yr College Enroll 1 year | 4yr College Enroll Full Time 1 year |
|---|---|---|---|---|---|---|---|---|---|
| **Self Reported Grit** | 2.663*** [0.244] | 0.015 [0.068] | 0.079** [0.112] | -19.963 [-0.053] | 6.808** [0.887] | 8.324*** [0.121] | 0.088*** [0.131] | 0.065** [0.093] | 0.067** [0.097] |
| | (0.451) | (0.010) | (0.031) | (21.732) | (3.041) | (2.852) | (0.030) | (0.031) | (0.031) |
| *Adj R-squared* | 0.262 | 0.080 | 0.120 | 0.257 | 0.346 | 0.293 | 0.089 | 0.133 | 0.134 |
| **Self Reported Locus of Control** | 1.486*** [0.142] | 0.024* [0.112] | 0.056* [0.084] | -7.319 [-0.021] | 6.717** [0.091] | 3.001 [0.045] | 0.094*** [0.147] | 0.051* [0.076] | 0.031 [0.047] |
| | (0.446) | (0.010) | (0.030) | (19.780) | (2.922) | (2.817) | (0.029) | (0.030) | (0.030) |
| *Adj R-squared* | 0.223 | 0.088 | 0.115 | 0.255 | 0.347 | 0.280 | 0.093 | 0.130 | 0.126 |
| **Self-Control Work** | 2.792*** [0.252] | 0.013 [0.058] | 0.102*** [0.144] | -33.992 [-0.089] | 1.129 [0.014] | 4.178 [0.060] | 0.091*** [0.134] | 0.063** [0.089] | 0.047 [0.067] |
| | (0.456) | (0.010) | (0.031) | (21.531) | (3.157) | (2.956) | '(0.031) | (0.031) | (0.031) |
| *Adj R-squared* | 0.267 | 0.079 | 0.128 | 0.262 | 0.338 | 0.282 | 0.090 | 0.132 | 0.129 |
| **Self-Control Interpresonal** | 1.532*** [0.138] | 0.016 [0.068] | 0.112*** [0.157] | -39.898* [-0.102] | 6.624** [0.085] | 12.624*** [0.179] | 0.077** [0.0113] | 0.079** [0.111] | 0.052* [0.074] |
| | (0.477) | (0.010) | (0.032) | (21.924) | (3.112) | (2.907) | (0.031) | (0.032) | (0.031) |
| *Adj R-squared* | 0.222 | 0.080 | 0.132 | 0.265 | 0.345 | 0.310 | 0.085 | 0.136 | 0.130 |

Standardized coefficients in brackets. Standard errors of estimated coefficients in parenthesis. Additional controls included in the model are: KBIT Scaled Score, Age, Ethnicity, Gender, FRL, SPED, ELL and household income.

* Indicates P-values<0.1

** Indicates P-values<0.05, and

*** Indicates P-values<0.01.

## Discussion and conclusions

Using data from a study of high school seniors (N = 513), we considered the potential of survey effort measures as proxy measures of character traits. Surveys often resemble routine paperwork and tasks that people have to complete in their everyday lives. For students, in particular, surveys completed at school can resemble schoolwork or homework. Therefore, we hypothesized that measuring the effort students put into surveys can provide relevant information about their grit and self-control, two character traits that correlate with academic and life success.

Two survey effort measures have shown recent promise: item non-response and careless answering. We contribute to previous research in two ways. First, we complement the work of Hitt, Trivitt and Cheng [12], and Hitt [6] on the validity of survey effort measures in adolescents by studying their correlation with teacher reports of students' skills, academic outcomes at the end of high school, and college attendance. Secondly, we looked at the relationship between survey effort measures and performance task measures of academic diligence and frustration tolerance.

Our results showed the promise of survey effort measures when used as proxy measures of grit and self-control. Both item non-response and careless answering showed convergent validity via negative correlations with self-reported and teacher-reported measures of grit and self-control. Although we acknowledge that the magnitudes of the correlations between survey effort and survey self-reported measures appear small and more replication of these results is needed, they are at least as large as the correlations reported in prior literature validating other behavioral-task measures of traits related to grit, and self-control. Item non-response demonstrated criterion validity through significant negative correlations with high school GPA, the probability of attempting to take the SAT, SAT scores, performance on end-of-senior-year

math and reading tests, and the probability of being enrolled in college one year after graduation. Careless answering also showed significant correlations with senior year GPA, attempting the SAT, end-of-senior-year math and reading test scores, and college enrollment. We acknowledge that one of our outcome measures–high-school graduation–had limited variability, suggesting a restriction of range concern. That said, this was not a concern for the other eight academic outcome measures.

We note one key limitation of our study: we only used a convenience sample of high school students in the United States. We encourage further replication work using other samples and settings to corroborate our results.

We believe that this study adds evidence to the potential of survey effort measures to provide meaningful information about students' character traits related to grit and self-control. These measures provide researchers and evaluators with a relatively easy source of information on students' traits related to grit and self-control in a manner that is not affected by biases that can affect self-reported or teacher-reported measures as respondents are usually unaware they are being monitored on their survey effort. In addition, they open the opportunity to gain further insights on character traits using previously collected data that had no direct measures of these skills [25]. We acknowledge, however, that these measures could also be biased and manipulated if used in higher stakes educational decisions or if students become aware of the fact that their survey behavior is being observed.

## Supporting information

**S1 Appendix.**
(DOCX)

## Acknowledgments

We would like to thank Julie Trivitt for her help in the early stages of this paper and Albert Cheng and Collin Hitt for their comments and feedback on our results. We also thank conference participants at the 42[nd] AEFP Annual Conference and the University of Arkansas Department of Education Reform Brownbag Seminar Series for all their feedback. Any errors are our own.

## Author Contributions

**Conceptualization:** Gema Zamarro, Angela L. Duckworth, Sidney K. D'Mello.

**Data curation:** Angela L. Duckworth, Sidney K. D'Mello.

**Formal analysis:** Gema Zamarro, Malachi Nichols.

**Writing – original draft:** Gema Zamarro, Malachi Nichols, Angela L. Duckworth, Sidney K. D'Mello.

**Writing – review & editing:** Gema Zamarro, Malachi Nichols, Angela L. Duckworth, Sidney K. D'Mello.

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
