## [Decision Letter · Decision Letter 0]

24 Dec 2019

PONE-D-19-29452

Further Validation of Survey Effort Measures of Relevant Character Skills: Results from a Sample of High School Students

PLOS ONE

Dear Dr. Zamarro,

Thank you for submitting your manuscript to PLOS ONE. After careful consideration, we feel that it has merit but does not fully meet PLOS ONE’s publication criteria as it currently stands. Therefore, we invite you to submit a revised version of the manuscript that addresses the points raised during the review process.

We would appreciate receiving your revised manuscript by Feb 07 2020 11:59PM. To enhance the reproducibility of your results, we recommend that if applicable you deposit your laboratory protocols in protocols.io, where a protocol can be assigned its own identifier (DOI) such that it can be cited independently in the future. For instructions see: http://journals.plos.org/plosone/s/submission-guidelines#loc-laboratory-protocols

We look forward to receiving your revised manuscript.

Kind regards,

Frantisek Sudzina

Academic Editor

PLOS ONE

2. Please provide additional details regarding participant consent.

In the ethics statement in the Methods and online submission information, please ensure that you have specified (i) whether consent was informed and (ii) what type you obtained (for instance, written or verbal, and if verbal, how it was documented and witnessed).

If your study included minors, state whether you obtained consent from parents or guardians.

If the need for consent was waived by the ethics committee, please include this information.

Reviewers' comments:

Reviewer's Responses to Questions

**Comments to the Author**

1. Is the manuscript technically sound, and do the data support the conclusions?

Reviewer #1: Partly

Reviewer #2: Partly

2. Has the statistical analysis been performed appropriately and rigorously? 

Reviewer #1: Yes

Reviewer #2: Yes

3. Have the authors made all data underlying the findings in their manuscript fully available?

Reviewer #1: No

Reviewer #2: No

4. Is the manuscript presented in an intelligible fashion and written in standard English?

Reviewer #1: No

Reviewer #2: No

5. Review Comments to the Author

Reviewer #1: 1. The author's do not sufficiently acknowledge that their small effect sizes should be interpreted with caution.

2. The authors seem to conduct the appropriate analyses (partial correlations, regression) to answer their research questions.

3. The data were not made public, however the authors do acknowledge in the Data Availability Statement (not in the body of the manuscript itself) that they cannot make the data open due to IRB restrictions.

4. This manuscript contained several grammatical, sentence structure, and APA formatting errors, all of which significantly hindered comprehension of its content.

Reviewer #2: Thank you for the opportunity to review this interesting research. This article aims to further validate survey effort measures as potential proxy measures for characteristics like grit and self-control. Strengths include integrating multiple sources of information (self-reported information, teacher reports, performance task measures, and administrative records). The concept of effort spent on tasks as a proxy for different personality or character traits is an interesting premise.

My greatest concerns are about the details that are provided, the language used, and consistency throughout the manuscript, which make it unclear what was done and how that resulted in the results. Below are comments related to the main criteria, which I hope can be useful to the authors in considering how to refine their work and improve the contribution to the literature.

1. While the article is written in standard English, the aims and contributions are inconsistently presented. Some parts provide extra details that seem irrelevant, and then insufficient detail is provided in other parts. As a whole, it’s not clear from the narrative what this study shows and how it contributes to existing knowledge (that information is there, but the focus and use of terms is inconsistent, adding confusion). Careful consideration of the structure, how details can be clearly yet concisely conveyed, and consistency across parts of the narrative would be beneficial. In addition, there are numerous grammatical errors throughout, suggesting a lack of care (that is a bit amusing with a paper on aspects related to conscientiousness).

2. In terms of sharing data, while it's understandable that data cannot be released, it would be useful to include the code files, if possible. In addition, for each of the scales that the careless answering is calculated on, indicate the number of items included in those scales and reliability information, which speaks to the extent to which inconsistencies might be due to the person versus to the scale. It would also be helpful to have a supplemental file that includes the extent to which a particular item is unpredictable.

3. The introduction immediately assumes that the reader views conscientiousness as a character skill. That is debatable, as most uses of the word clearly places it in the personality space, but not necessarily as a skill. Grit is defined here as persistence in long-term tasks, but generally this has involved not only persistence but also passion. Rationale for using these terms as stated would be useful to provide context to the reader. And then in terms of use, this seems to be using these terms more from a personality than a character skills perspective, so rationale for taking this lens (and what is meant by this lens) is needed. In addition, clear definition of terms, such as survey effort measures, parametrization, careless answering.

4. Method:

a) p. 4 notes that data were collected on as many students as resources allowed. Meaning what? How many were included? What were the resources here? While pointing to Meindl et al., 2019, what is the reader supposed to see that citation for? Greater specificity about the students involved would be useful. This vaguely notes 513 high school seniors from a public school. Some indication of the socioeconomic and ethnic makeup of students would be useful.

b) What is the National Student Clearinghouse? Provide a citation or website.

c) In describing what students completed, indicate the exact number of items, not “about 100” and “about 10 more final questions”.

d) When did the sessions occur? Was this during school? In class or outside of class? Did all students in a class complete, or only some? When was the second session? How close in time? How many students did teachers report on? The description of this comes across as quite vague and hard to make sense of what was really done by whom.

e) What is meant by the number of answerable questions to which a student should have responded to? Without knowing the measures being used, not clear what is meant by this.

f) Figure 1 is hard to read, with the large percentage of complete cases. It might be useful to break this into two parts, one indicating response or not, then indicate the distribution of the 53% with missing responses (with the axis adjusted accordingly.

g) In describing careless answering, a reliable scale will not necessarily be consistent (at least in the psychological sciences), as the items can only approximate the underlying construct. While on average across a sample there is consistency, for any individual a variety of factors can cause an item to be less inconsistent. It’s a strong assumption to say that variance is due to carelessness (i.e., a problem with the person), and not due, for instance to the wording of the question or the person carefully discriminating between two option choices.

h) p. 7, noting the estimated correlation coefficient, is this the Pearson r? Say this directly. And be careful about making interpretations about what the correlation does or does not indicate that students do.

i) Did teachers complete the 6 and 8 items for grit and self-control for each student, or did they read through the items and make a judgment call about the extent to which those represent the student?

j) For the questions being asked of teachers, it seems odd that a homeroom teacher is assessing this, as I would think they would have less experience redirecting attention and determining homework completed, (for readers less aware of how the school structure works). Rationale for choosing the 3 subjects would be useful.

k) I’m not clear why details on the ADT correlations in a different study are reported in the text, whereas similar information is not included for all the other measures.

l) Spell out acronyms on first use (GPA, SAT).

m) I find it hard to follow the tests that are planned and the expectations. A table or figure could be a useful way to convey the analytic strategy.

5. Results:

a) Information about the participants are finally provided in the results. This would be useful much earlier. Tables should also be numbered in the order they are mentioned in the text (Table 2 is noted before Table 1). The description of the table could be briefer to be less repetitious with the table (or the info could be descriptively given in the text and then just summarise the measures in the table). The description of the measure responses could also be more clear and concise.

b) Consider combining tables 3 and 4, so it’s easier to directly compare the direct and partial correlations.

c) For comparison, it would be useful to also predict the outcomes with the self-report and teacher reported measures, to see if a similar pattern to the frustration measures occur. This is especially necessary as the conclusion indicates that they are tested as a proxy measure of character skills – so need to directly see that they are capturing the same thing.

6. Conclusions

a) The conclusion notes that this can give relevant information about conscientiousness. But the focus is on grit and self-control, not on conscientiousness (though that is measured in the Big 5) – if that’s the goal, then should be addressed directly (and also have more consideration of how the different self, teacher, and performance measures intersect with conscientiousness. (Though the third paragraph then instead speaks of skills related to conscientiousness. It would be useful to be consistent in the narrative throughout.)

b) The overlap with both the self-reported measures and teacher measures are quite small, suggesting they are not a particularly good proxy (perhaps are capturing different variance). Some discussion of this should be included.

c) At some point should discuss the assumptions that are being made with the survey effort measures. The last paragraph suggests these are not affected by biases that affect self and teacher reported measures, but instead they reflect biases and assumption of the researchers, which should be explicitly acknowledged.

6. PLOS authors have the option to publish the peer review history of their article (what does this mean?). If published, this will include your full peer review and any attached files.

Reviewer #1: No

Reviewer #2: No

---

## [Author Response · Author response to Decision Letter 0]

16 Jun 2020

RESPONSE TO COMMENTS FROM REVIEWER 1

1) The author's do not sufficiently acknowledge that their small effect sizes should be interpreted with caution.

This is now better acknowledged in the text both in the introduction and conclusion of the paper. Please, see pages 3 and 4. 

2) The data were not made public, however the authors do acknowledge in the Data Availability Statement (not in the body of the manuscript itself) that they cannot make the data open due to IRB restrictions.

All data collected in this study is considered confidential. After further studying and discussing our IRB at the University of Pennsylvania we learned that our IRB protocol only allows sharing a fully de-identified dataset with collaborators for research analysis directly involved in this project after receiving approval by the PI of the project, who assumed responsibility for how and where the data will be stored and analyzed. The reason being that this research involves minors and this confidentially was promised to participants. Therefore, we regret that our IRB does not allow us to share the data. We hope you understand given the nature of the data and the fact that our research involves minors.

3) This manuscript contained several grammatical, sentence structure, and APA formatting errors, all of which significantly hindered comprehension of its content. This manuscript includes several grammatical errors and would benefit substantially from thorough copy editing. Moreover, the organization of this manuscript (e.g., the subhead “Data” should be “Materials and Methods”) does not fit with APA guidelines and that of PLOS (https://journals.plos.org/plosone/s/submission-guidelines). Finally, as per APA guidelines, please report measures and results in past tense. These errors and issues limit the manuscripts readability quite a bit.

We have revised the text to follow APA style and the text has also been copy-edited for grammatical errors.

4) Use of the term “character skills” to describe the personality trait conscientiousness and its facets is odd. Decades of research has identified conscientiousness as a personality trait. Please include rationale for why conscientiousness and its facets are termed as a character skill and not a personality trait in this context. Alternatively, given that conscientiousness is not measured directly, I suggest focusing on grit and self-control throughout the manuscript, especially the methods and results sections. 

There is a lack of consensus across disciplines on the use of these terms and hence the confusion as our research team is multidisciplinary. Economists use the term non-cognitive skills while education policy researchers refer to these terms as character skills. We have revised the text to use terminology more in line with the psychology literature.

5) Relatedly, you introduce the manuscript’s focus on validation by stating that “researchers have struggled to find valid measures of these skills with many existing datasets lacking any measures at all”. In fact, measures to assess trait level conscientiousness have been very well validated (see BFI, BFI2, NEO, etc.) If you decide to focus on conscientiousness and its facets, this sentence is problematic.

Available self-reported measures, despite being validated, can be problematic when used to evaluate the effects of education policies or interventions (see Duckworth and Yeagar, 2015), that is the point we were trying to make with this sentence. We have revised the text to try to make this more clear. See the revised abstract and the first paragraph of the Introduction page 1.

6) Please include reference to the measures you used as well as example items for each.

References to the measures and example items are included in the text. We have also added Table A.1 to the appendix which describes all the scales included in our careless answering measures.

7)In the Survey Effort Measures section, please clarify what constitutes as “legitimate skips” with regards to item non-responding.

These are questions that, given a previous student’s responses, he/she is not required to complete in the survey. We have tried to clarify now this more in the text. See pages 6 and 7.

8) Please clarify why teachers rated the grit measure items as a composite instead of each item individually. This approach limits the measures’ predictive power as well as its ability to assess the measures’ reliability. Also, please report the number of students each teacher gave ratings for.

Teachers provided an overall rating on their student’s levels of grit and self-control to minimize burden, as teachers reported on all of their current students participating in the study. This measurement approach using a single overall assessment of personality traits has been proved to show adequate levels of convergent and discriminatory validity, test-retest reliability, and convergent validity between self-and observed-ratings (See, Gosling, Rentfrow, & Swann, 2003; Rammstedt & John, 2007)

9)Clarify what including self-report measures “for completeness” means in the context of this study.

The point we wanted to make here was that, even though others have already studied the relationship between survey effort measures and self-reports, we also study this relationship as it provides evidence of convergent vaidity. We have rewritten this sentence on page 10 and eliminated “for completeness”.

10) Please explain how and why the measures for grit and self-control were adapted from the original scales.

Most studies of grit include at least 8 items. For the sake of brevity, we followed Duckworth and Quinn (2009) and used only the 5 items below:

1) I finish whatever I begin.

2) I work independently with focus.

3) I tried very hard even after experiencing failure.

4) I stay committed to my goals. 

5) I keep working hard even when I feel like quitting

Source: Duckworth, A. L., & Quinn, P. D. (2009). Development and validation of the short grit scale (Grit-S). Journal of Personality Assessment, 91, 166-174. 

Similarly, we included the following 8 items from the Brief Self-Control Scale to get at the overarching construct of self-control.

Self-control at work:

1) I come to class prepared.

2) I pay attention and resist distractions in class.

3) I remember and follow directions.

4) I get to work right away, instead of waiting until the last minute.

Self-Control (Interpersonal)

5) I allow others to speak without interruption.

6) I am polite to adults and classmates.

7) I can control my temper.

8) I can remain calm even when criticized or otherwise provoked.

Source: Tangney, J. P., Baumeister, R. F., & Boone, A. L. (2004). High self-control predicts good adjustment, less pathology, better grades, and interpersonal success. Journal of Personality, 72, 271-322.

Overall, both modified scales continued to capture the underlying measures as intended. 

11) Its my understanding that GPA was on a scale from 1.0-4.0. Please explain why your measure of GPA is on a scale from 1 to 100.

In the state where our data comes from, high schools vary on their calculations and scale used for their GPAs. Therefore, we converted GPA to a 100-point scale with the help of district-provided handbooks and information from the College Board. 

12) Please report effect sizes of regression models throughout results sections instead of or in addition to standard deviation increases or decreases.

We have added estimated coefficients and standard errors of estimated coefficients in addition to standardized coefficients to the current regression result tables. See the new Table 4.

13) Please include a note for Table 1 explaining what ELL, SPED etc. mean. 

Done

14) In the discussion section, address limitations for the lack of variation in outcome measures (e.g., graduating high school). Relatedly, please address the limitations of your item non-response measure using (1) a continuous measure with 47% of participants’ completing all questions (2) a dichotomous measure of non-responding is problematic (i.e., not answering 1 question is not the same as not answering 10 questions), as well as the implications of the non-normal distribution for this measure (e.g., that careless responding may be a better predictor).

In the discussion section, we now acknowledge that one of our outcomes measures – high-school graduation – had limited variability, suggesting a restriction of range concern. That said, this was not a concern for the other eight academic outcome measures.

15) While you do recognize the small effect sizes (i.e., “Although the magnitudes of the correlations between survey effort and survey self-reported measures may appear small, they are at least as large as the correlations reported in prior literature validating other behavioral-task measures of conscientiousness, grit, and self-control.”), please address this limitation in the discussion section and caution readers in over-interpreting small effect sizes. Specifically, we may trust some effects more than others. Likewise, some effects may require replication more than others. 

This is now better acknowledged in the discussion section.

RESPONSE TO COMMENTS FROM REVIEWER 2

1) While the article is written in standard English, the aims and contributions are inconsistently presented. Some parts provide extra details that seem irrelevant, and then insufficient detail is provided in other parts. As a whole, it’s not clear from the narrative what this study shows and how it contributes to existing knowledge (that information is there, but the focus and use of terms is inconsistent, adding confusion). Careful consideration of the structure, how details can be clearly yet concisely conveyed, and consistency across parts of the narrative would be beneficial. In addition, there are numerous grammatical errors throughout, suggesting a lack of care (that is a bit amusing with a paper on aspects related to conscientiousness).

We have considerably revised the text to address these concerns. We have also used the services of a copy editor to eliminate any possible grammatical errors.

2) In terms of sharing data, while it's understandable that data cannot be released, it would be useful to include the code files, if possible. In addition, for each of the scales that the careless answering is calculated on, indicate the number of items included in those scales and reliability information, which speaks to the extent to which inconsistencies might be due to the person versus to the scale. It would also be helpful to have a supplemental file that includes the extent to which a particular item is unpredictable.

We have created the appendix table describing all this information. See Table A.1 page 33. In terms of the code, if the editor considers it necessary, we could document and provide the code if the paper is accepted for publication.

3) The introduction immediately assumes that the reader views conscientiousness as a character skill. That is debatable, as most uses of the word clearly places it in the personality space, but not necessarily as a skill. Grit is defined here as persistence in long-term tasks, but generally this has involved not only persistence but also passion. Rationale for using these terms as stated would be useful to provide context to the reader. And then in terms of use, this seems to be using these terms more from a personality than a character skills perspective, so rationale for taking this lens (and what is meant by this lens) is needed. In addition, clear definition of terms, such as survey effort measures, parametrization, careless answering.

Thank you for this comment. We acknowledge the fact that these terms might be confusing and adding to the confusion is the fact that different disciplines (e.g. economics, psychology, education policy) are using these terms in different ways. We have tried to clarify the definition of these terms in the text and tried to better justify how we are using them to avoid misunderstandings.

4) Method:

a) p. 4 notes that data were collected on as many students as resources allowed. Meaning what? How many were included? What were the resources here? While pointing to Meindl et al., 2019, what is the reader supposed to see that citation for? Greater specificity about the students involved would be useful. This vaguely notes 513 high school seniors from a public school. Some indication of the socioeconomic and ethnic makeup of students would be useful.

Following this comment, we have added more details about the sample. A description of the makeup of students in our sample can be found at the end of page 4.

b) What is the National Student Clearinghouse? Provide a citation or website.

We have described and added a link for the National Student Clearinghouse see page 4

c) In describing what students completed, indicate the exact number of items, not “about 100” and “about 10 more final questions”.

Different students would complete a different total number of questions because some questions are only required if they answer certain options in prior questions. That is why we did not give the exact number of questions as it would slightly vary by student.We changed this part of the text and eliminated the reference to the number of questions to avoid the confussion.

d) When did the sessions occur? Was this during school? In class or outside of class? Did all students in a class complete, or only some? When was the second session? How close in time? How many students did teachers report on? The description of this comes across as quite vague and hard to make sense of what was really done by whom.

Those students whose parents did not opt-out were surveyed in three sessions:

Session 1: Students were administered the assent forms and Mill Hill Vocabulary test during senior panning periods in school (37 minutes sessions). A large make-up session (n ~300) was held on the final day of testing in the library computer lab. This session occurred in November 2012.

Session 2: Students were administered the a Cognitive battery, Non-Cognitive battery, and the Academic Diligence task during one 2.5 hour session. Students were administered the tasks in either the cafeteria (50-200 students per day) or individual classrooms on Macbook laptop computers (~30 students/class per day). This session occurred in January 2013.

Session 3: Students were administered the Frustration Task during senior planning periods. Students were administered the task in the library on Macbook laptop computers. For four periods per day, two classes of students were tested during each period (approximately 30-60 students per period). This session occurred in May 2013. 

Teachers reported on all of their participating students in the study.

e) What is meant by the number of answerable questions to which a student should have responded to? Without knowing the measures being used, not clear what is meant by this.

This means all questions that a student is supposed to answer not counting as non-response legit skips because of survey routing. We have tried to clarify this better in the text. See pages 6 and 7.

f) Figure 1 is hard to read, with the large percentage of complete cases. It might be useful to break this into two parts, one indicating response or not, then indicate the distribution of the 53% with missing responses (with the axis adjusted accordingly.

Following the reviewer’s advice we have eliminated prior Figure 1 and opted for discussing the results in the text and added a new figure representing the distribution of item non-response rates for those with missing responses (new Figure 1).

g) In describing careless answering, a reliable scale will not necessarily be consistent (at least in the psychological sciences), as the items can only approximate the underlying construct. While on average across a sample there is consistency, for any individual a variety of factors can cause an item to be less inconsistent. It’s a strong assumption to say that variance is due to carelessness (i.e., a problem with the person), and not due, for instance to the wording of the question or the person carefully discriminating between two option choices.

We acknowledge that our survey effort measures, including careless answering, are proxy measures for student effort and as so, would have some noise. Previous literature has suggested their promise as proxy measures for personality traits related to grit and self-control (Hedengren and Stratmann, 2012; Hitt, Trivitt and Cheng 2016; Hitt, 2015; Zamarro et al., 2018). Therefore, we believe that despite being potentially noisy they can contain relevant information about an individual’s diligence which will relate to their levels of grit and self-control.

h) p. 7, noting the estimated correlation coefficient, is this the Pearson r? Say this directly. And be careful about making interpretations about what the correlation does or does not indicate that students do.

Yes, this is a Pearson’s correlation coefficient. We have now stated this in the text and rewrote this paragraph to not make any interpretations about what this correlation indicates about student behaviors.

i) Did teachers complete the 6 and 8 items for grit and self-control for each student, or did they read through the items and make a judgment call about the extent to which those represent the student?

To limit the workload on teachers, as they reported on all their participating students, teachers provided an overall rating on their student’s levels of grit and self-control to minimize their burden. This measurement approach using a single overall assessment of personality traits has been proved to show adequate levels of convergent and discriminatory validity, test-retest reliability, and convergence between self-and observed-ratings (See, Gosling, Rentfrow, & Swann, 2003; Rammstedt & John, 2007)

j) For the questions being asked of teachers, it seems odd that a homeroom teacher is assessing this, as I would think they would have less experience redirecting attention and determining homework completed, (for readers less aware of how the school structure works). Rationale for choosing the 3 subjects would be useful.

Homeroom, English, and social science teachers were asked to do so. We believe that these three teachers can provide meaningful information about students. Given our experience in the field, we also think that homeroom teachers are capable to report on these items. 

k) I’m not clear why details on the ADT correlations in a different study are reported in the text, whereas similar information is not included for all the other measures.

Details on correlations for the ADT and MTFT are provided for comparison with our estimated correlations for measures of survey effort. We believe this comparison is meaningful because these are direct task measures and we believe effort in a survey could be view as a behavioral task.

l) Spell out acronyms on first use (GPA, SAT).

This is now done

m) I find it hard to follow the tests that are planned and the expectations. A table or figure could be a useful way to convey the analytic strategy.

We have rewritten and reorganized the paper including our empirical strategy section, pages 15 and 16, and we hope this is now more clear in this version.

5) Results

a) Information about the participants are finally provided in the results. This would be useful much earlier. Tables should also be numbered in the order they are mentioned in the text (Table 2 is noted before Table 1). The description of the table could be briefer to be less repetitious with the table (or the info could be descriptively given in the text and then just summarise the measures in the table). The description of the measure responses could also be more clear and concise.

We have moved the information about participants earlier in the text and addressed the issue with the numbering of Tables. We have also revised the text aiming to increase clarity in describing the tables.

b) Consider combining tables 3 and 4, so it’s easier to directly compare the direct and partial correlations.

We have done this

c) For comparison, it would be useful to also predict the outcomes with the self-report and teacher reported measures, to see if a similar pattern to the frustration measures occur. This is especially necessary as the conclusion indicates that they are tested as a proxy measure of character skills – so need to directly see that they are capturing the same thing.

We have added new estimates predicting the outcomes with the self-report and teacher reported measures. See new Tables 5 and 6 that are described in the text in page 24.

6) Conclusions

a) The conclusion notes that this can give relevant information about conscientiousness. But the focus is on grit and self-control, not on conscientiousness (though that is measured in the Big 5) – if that’s the goal, then should be addressed directly (and also have more consideration of how the different self, teacher, and performance measures intersect with conscientiousness. (Though the third paragraph then instead speaks of skills related to conscientiousness. It would be useful to be consistent in the narrative throughout.)

We revised the text to be more consistent on our use of these terms and make it more clear to the reader.

b) The overlap with both the self-reported measures and teacher measures are quite small, suggesting they are not a particularly good proxy (perhaps are capturing different variance). Some discussion of this should be included.

As pointed out in the text our observed correlations are of the same magnitude than those observed with other designed behavioral tasks like the Diligence Task and the Frustration task and so, we disagree that these correlations are as small as this reviewer suggests. In any case, this is now better acknowledged in the conclusions where we point out the need for more replication.

c) At some point should discuss the assumptions that are being made with the survey effort measures. The last paragraph suggests these are not affected by biases that affect self and teacher reported measures, but instead they reflect biases and assumption of the researchers, which should be explicitly acknowledged.

We now better acknowledge in the conclusions section that our survey effort measures are not free of potential biases:

“We acknowledge, however, that these measures could also be biased and manipulated if used in higher stakes educational decisions or if students become aware of the fact that their survey behavior is being observed.”

---

## [Editor Report · Decision Letter 1]

16 Jun 2020

Validation of survey effort measures of grit and self-control in a sample of high school students

PONE-D-19-29452R1

Dear Dr. Zamarro,

We’re pleased to inform you that your manuscript has been judged scientifically suitable for publication and will be formally accepted for publication once it meets all outstanding technical requirements.

Kind regards,

Frantisek Sudzina

Academic Editor

PLOS ONE

---

## [Editor Report · Acceptance letter]

22 Jun 2020

PONE-D-19-29452R1 

Validation of survey effort measures of grit and self-control in a sample of high school students 

Dear Dr. Zamarro:

I'm pleased to inform you that your manuscript has been deemed suitable for publication in PLOS ONE. Congratulations! Your manuscript is now with our production department. 

Kind regards, 

on behalf of

Dr. Frantisek Sudzina 

Academic Editor

PLOS ONE